# Computational Investigation of Mechanisms for pH Modulation of Human Chloride Channels

**DOI:** 10.3390/molecules28155753

**Published:** 2023-07-30

**Authors:** Kathleen Elverson, Sally Freeman, Forbes Manson, Jim Warwicker

**Affiliations:** 1Division of Evolution, Infection and Genomics, Faculty of Biology, Medicine and Health, The University of Manchester, Manchester M13 9PT, UK; 2Division of Pharmacy and Optometry, School of Health Sciences, Faculty of Biology, Medicine and Health, The University of Manchester, Manchester M13 9PT, UK; 3Division of Molecular and Cellular Function, Faculty of Biology, Medicine and Health, Manchester Institute of Biotechnology, The University of Manchester, Manchester M1 7DN, UK

**Keywords:** bestrophin, chloride channels, calcium binding, pH dependence, protein electrostatics

## Abstract

Many transmembrane proteins are modulated by intracellular or extracellular pH. Investigation of pH dependence generally proceeds by mutagenesis of a wide set of amino acids, guided by properties such as amino-acid conservation and structure. Prediction of pKas can streamline this process, allowing rapid and effective identification of amino acids of interest with respect to pH dependence. Commencing with the calcium-activated chloride channel bestrophin 1, the carboxylate ligand structure around calcium sites relaxes in the absence of calcium, consistent with a measured lack of pH dependence. By contrast, less relaxation in the absence of calcium in TMEM16A, and maintenance of elevated carboxylate sidechain pKas, is suggested to give rise to pH-dependent chloride channel activity. This hypothesis, modulation of calcium/proton coupling and pH-dependent activity through the extent of structural relaxation, is shown to apply to the well-characterised cytosolic proteins calmodulin (pH-independent) and calbindin D9k (pH-dependent). Further application of destabilised, ionisable charge sites, or electrostatic frustration, is made to other human chloride channels (that are not calcium-activated), ClC-2, GABA_A_, and GlyR. Experimentally determined sites of pH modulation are readily identified. Structure-based tools for pKa prediction are freely available, allowing users to focus on mutagenesis studies, construct hypothetical proton pathways, and derive hypotheses such as the model for control of pH-dependent calcium activation through structural flexibility. Predicting altered pH dependence for mutations in ion channel disorders can support experimentation and, ultimately, clinical intervention.

## 1. Introduction

The modulation of transmembrane protein function by pH is an important feature of many systems, in particular considering the role of pH gradients in cell physiology. Computational methods aimed at predicting pH dependence can aid the understanding of pH-dependent processes, but testing is required to realize their potential. Here, the molecular bases of pH dependence are investigated for human chloride channels, comparing reported experimental data with structure-based calculations, thus addressing the twin questions of model accuracy and molecular mechanism.

Bestrophin 1 (Best1) is a Ca^2+^-dependent Cl^−^ channel that localises to the basolateral plasma membrane of retinal pigment epithelium (RPE) cells [1,2]. Mutations in the *BEST1* gene lead to a set of inherited retinal dystrophies (IRDs) called bestrophinopathies, caused by protein instability and loss of function of the Best1 protein. Bestrophinopathies can cause severe vision deterioration in patients and are incurable.

Structures of chicken Best1 (cBest1) and a bacterial homologue (*Klebsiella pneumonia*, KpBest) have been reported [3,4,5,6,7,8,9]. The barrel-shaped channel is a homopentamer comprising five subunits arranged around a central pore, with each subunit composed primarily of α-helices. cBest1 possesses five symmetric calcium clasp sites that assemble together to form a belt around the centre of the channel [6]. A series of cryo-electron microscopy (cryo-EM) cBest1 structures revealed the mechanism of a Ca^2+^-dependent gating cycle [9]. When unoccupied by Ca^2+^, the calcium clasps are mostly disordered, and there is a degree of conformational flexibility between the transmembrane and cytosolic regions. The binding of Ca^2+^ ions rigidifies the clasps and the channel equilibrates between two conformations of similar energy; Ca^2+^ bound/open and Ca^2+^ bound/closed. The open conformation allows permeation of hydrated anions [9].

Bovine bestrophin 2 (bBest2) structures have also been reported, with and without calcium bound, and show a similar overall architecture to KpBest and cBest1 [10]. However, the ion selectivity of the channel is controlled by different residues and the conductance of smaller ions, including Cl^−^, can occur in the absence of calcium [10]. Interestingly, the theme of calcium-dependent conformation of the calcium clasp is maintained in Best2, although a major disordering is evident in the absence of calcium for cBest1, bBest2 reveals an altered but not entirely disordered conformation when calcium is not bound. Best vitelliform macular dystrophy (BVMD) and autosomal recessive bestrophinopathy (ARB) are two of the clinically distinct diseases associated with mutations in *BEST1* [11]. BVMD mutants are clustered around important functional sites for the channel including the calcium clasp, whilst ARB mutants are located mainly outside these regions [12].

Some systems that bind and respond to calcium are also sensitive to pH, with examples in the aqueous phase and membrane proteins [13,14]. Due to the difficulty of isolating the relevant calcium-bound and unbound structures, as well as identifying specific sites of proton binding, a detailed molecular understanding of pH coupling to calcium binding is lacking. It is thought that the close proximity of carboxylate groups from the sidechains of aspartic and glutamic acid sidechains increases the pKas of these groups up towards neutral pH when calcium is not bound if the site remains in its calcium binding configuration. Another example is the family of acid-sensing ion channels (ASICs) [15]. Calcium is a channel block and protons cause channel opening. Proton dependence of ASIC activity has been reviewed [16] and likely depends on several factors, including histidine residues. One of these factors, demonstrated in rat ASIC3, is a ring of glutamates that mediates both proton and calcium sensitivity [17]. Glutamic acid modulated pH dependence around neutral pH implies a substantial pKa shift (ΔpKa) from the normal pKa value (4.4). Such a scale of ΔpKa for amino acid sidechain carboxylates can be observed in enzyme active sites and other systems [18]. Mechanistically, the large and energetically unfavourable ΔpKa is achieved through maintaining a relatively rigid conformation in an environment that is at least partially buried from solvent water that would hydrate and stabilise the ionised form. This degree of burial can be achieved in calcium binding sites, and the proximity of carboxylates to each other (at least two are required to neutralise the calcium ion charge) also contributes to positive ΔpKa. It is, therefore, unsurprising that calcium binding sites with carboxylate ligands could also act as pH sensors at physiological pH, with calcium and proton ions competing for the same site. More generally, the term electrostatic frustration has been applied to ionisable groups for which ionisation is energetically unfavourable in a buried environment, but would be favourable when solvent exposed [19]. This observation has been generalised on the Protein-Sol web server [20], where calculations of pKa for ionisable groups are colour-coded for predictions of stabilisation or destabilisation (electrostatic frustration). Thus, destabilised aspartic and glutamic acid carboxylate sidechains in buried environments, that are not sufficiently complemented by charge interactions, will have upward pKa shifts (from solvent exposed model compound values). Destabilised histidine imidazole groups will have downward pKa shifts. In each case the red colouring on the server tracks the predicted degree of electrostatic frustration.

In cases such as ASICs, the function is associated with the coupling of calcium and proton binding, but in other cases, it may be that modulation by calcium does not couple to proton sensitivity. An obvious mechanism to uncouple the two cation effects is to engineer substantial relaxation of the binding site, such that in the absence of calcium, carboxylate ligands would be distanced from each other. Then, with higher solvent exposure and stabilisation of ionised forms, the associated pKas would be closer to normal values and well below neutral pH. In this regard, it is intriguing that Best1 shows a high degree of structural relaxation in the absence of calcium binding and that it has been reported to be functionally pH-independent (with respect to extracellular and intracellular pH) at around neutral pH [21].

In this report, we first study the calcium clasp of Best1, investigating the carboxylate pKas in the calcium binding site, using established computational methods for pKa predictions [20,22], and rationalise the observed lack of pH sensitivity for Best1. Next, we assess the general principles of modulating the coupling of proton and calcium binding, with reference to well-characterised cytosolic proteins [13] as well as another chloride channel, TMEM16A. Finally, we expand the study of pH dependence to other human chloride channels. Generally, and unlike Best1, pH dependence of function is present and can be explained on the basis of specific ionisable amino acid sidechains that are in buried or relatively buried environments and experience electrostatic frustration. Best1 is notable for possessing a mechanism (the calcium clasp) that, rather than being adapted to add pH dependence, has evolved structural flexibility that, in turn, yields pH independence of function.

## 2. Results

### 2.1. Coupling of Calcium Binding and pH in Bestrophin 1

With Ca^2+^ bound to cBest1 (Figure 1A,B), calculation with the Protein-Sol pKa web tool gives predicted substantial destabilising ΔpKas (for that conformation but in the absence of calcium ion) of 4.0/D301 (pKa 8.0), >5/D303 (pKa > 9), >5/D304 (pKa > 9), and with a moderate stabilising ΔpKa of −1.5 for D302 (pKa 2.5). Note that pKa changes (and colour ramping) in the Protein-Sol display are capped at a magnitude of 5, sufficient to move aspartic and glutamic acid pKas up beyond neutral pH. When unoccupied by Ca^2+^, cBest1 Ca^2+^ clasps are disordered, with density for the surrounding residues missing in the structures [9] (Figure 1C). Therefore, the predicted electrostatic frustration that would exist in the apo state if the protein scaffold that binds calcium were maintained, is in practice relieved by disordering. If this disordering were not the case, then elevated aspartic acid sidechain pKas would lead to a pH dependence at physiological pH, whereby increasing proton concentration (lower pH) would compete with calcium for binding.

In the bBest2 (apo) structure without calcium, rather than complete disordering of the clasp, there is a partial loss of order for sidechains in this region (Figure 2). Calculated ΔpKas in the holo structure, -but in calculation without the calcium ion, (Figure 2B) are destabilising 3.3/D301, 4.9/D303, >5/D304, and stabilising 1.8/D302, and for the apo structure (Figure 2C) are destabilising 3.0/D304, and stabilising 0.2/D301, 0.5/D302, 0.1/D303. Therefore, the structural relaxation leads to a lowering of electrostatic frustration (positive carboxylate group ΔpKas) for Ca^2+^ binding residues overall in the bBest2 apo structure compared to the holo structure. Unlike cBest1 though, the predicted elevated pKa for D304 on apo hBest2 implies that there could be some pH dependence of calcium binding.

For the bestrophin case, a comparison has been made with pKa calculations when the pentameric protein is embedded in the membrane, using final snapshots from simulations available in MemProtMB [23]. Calcium-free cBest1 (6n26) and calcium-bound cBest1 (6n27) derived from this resource were simulated in membranes of zwitterionic dipalmitoyl phosphatidylcholine. Calcium was removed from the calcium-bound structure for pKa calculations. Summed over the acidic ligands around one calcium site, net protonation is zero for calcium-free (with only E300 ordered) and 2.0 for the site in its calcium-bound form, mirroring the predicted changes with bestrophins prior to molecular dynamics simulation. These predicted protonations are the same whether the simulated protein is extracted from the membrane or embedded within it. Since the membrane has little effect on electrostatic interactions at the calcium clasp, and since each calcium site is closer to the membrane in this system than in the other membrane proteins of this study, subsequent calculations are made without the membrane. Additionally, for bestrophin, pKa calculations were made with an alternate method, propka [24]. For 6n27, with calcium removed from the structure, elevated pKas are calculated for D303 (7.34) and D304 (5.80), qualitatively matching FDDH predictions of enhanced electrostatic interactions in the notional unrelaxed, ordered state.

The predictions in respect of electrostatic relaxation, particularly for cBest1, may provide an explanation for the measured pH independence of hBest1 activity around physiological pH [25]. In the absence of other interactions to stabilise the calcium-binding clasp, electrostatic frustration (from desolvation and repulsive charge−charge interactions) leads to an order to disorder transition so that elevated carboxylate sidechain pKas and pH dependence around neutral pH are avoided. Conversely, sufficient stabilising interactions could (in principle) be introduced around a calcium-binding site, such that its geometry would be maintained in the absence of calcium, and then pH dependence at neutral pH would be mediated by the carboxylate ligands of calcium. In either case (order to disorder or maintained order), it should be noted that the region of protein impacting on function of the calcium-binding site will include the site itself and at least the immediate surround of nearest neighbour amino acids, including (for Best1) the N-terminal region that abuts the site. It is, therefore, unsurprising that a set of BVMD loss-of-function, disease-associated mutations clusters around the belt of calcium clasps in the hBest1 oligomer [12].

### 2.2. Conformational Relaxation and pH Dependence of Calmodulin and Calbindin D_9k_

Some of the most well-studied calcium binding sites belong to the calmodulin superfamily of cytosolic proteins, containing helix-loop-helix modules (EF-hands). Although not transmembrane proteins, this superfamily is included due to the wealth of available data and facilitating a comparison between membrane and aqueous soluble proteins. Two prominent members of the superfamily, calmodulin (CaM) and calbindin D_9k_ (CaBP-9k), are displayed in Figure 3. Upon calcium binding, a large conformational change is observed for the globular domains of CaM [26,27]. At the calcium-binding site indicated (Figure 3A,B), it is predicted that conformational change leads to a relaxation of the electrostatic frustration that E31 would experience if held in the holo protein conformation but in the absence of calcium. From the pKa calculations (predicted destabilising ΔpKa for E31 of 4.9 in the holo form falling to 0.5 in the apo form), and in line with observed structural relaxation, it is predicted that calcium binding does not couple to pH (around neutral pH) for CaM. This suggestion is consistent with a reported pH independence of calcium binding to CaM in the range of 6.4−8.3 [28,29].

In comparison, CaBP-9k shows less structural rearrangement upon Ca^2+^ binding [32]. It has been suggested that preorganization of the Ca^2+^-binding site and a high Ca^2+^-binding affinity may be necessary for its buffering role [32]. Consistent with less structural change, predicted destabilising ΔpKas are present in both holo and apo forms of CaBP-9k (destabilising ΔpKa > 5 for E65 in both forms, Figure 3C,D). Elevated pKas for acidic residues in the Ca^2+^ binding site of apo CaBP-9k have been reported [33] together with a large decrease in calcium-binding affinity below pH 7.

Our hypothesis, that the extent of disordering and/or structural rearrangement in the absence of calcium binding modulates pH dependence, was developed from an analysis of structures around the calcium clasp of the bestrophin chloride ion membrane channels. It appears to transfer also to variation within the calmodulin superfamily, the most widely studied set of calcium-binding proteins.

### 2.3. pH Dependence of the Calcium-Activated Chloride Channel TMEM16A

Having used the Protein-Sol pka web tool to study the molecular basis of pH dependence and pH independence of calcium binding in bestrophins, we sought to apply the same method to study pH-dependent activity for representatives of other human chloride channel families. These have been broadly classed into bestrophins, anoctamins (ANO), cystic fibrosis transmembrane conductance regulator (CFTR), ligand-gated chloride channels (including GABA_A_ and glycine receptors), and voltage-gated chloride channels (ClCs) [34].

TMEM16A is a member of the anoctamin family expressed in a wide range of tissues and involved in secretion, intestinal motility, and contraction of smooth muscle cells [35], and (like Best1) is a calcium-activated chloride channel (CACC). It forms a homodimer with a hydrophilic ion-conducting pore [36,37]. Calcium binding to sites rich in acidic residues triggers conformational change that leads to pore opening [37,38]. Calculations of pKas were made around the calcium-binding sites of two pairs of calcium-bound/calcium-free structures of TMEM16A (5oyb/5oyg [37], and 7b5e/7b5d [38]). Of the 5 glutamic and aspartic acid sidechains that ligand calcium ions (E654, E702, E705, E734, D738), substantially elevated pKas are predicted for just 1 in each of the 4 structures (E734, destabilising ΔpKa between 1 and 2, in all but 7b5d, for which D738 has a predicted destabilising ΔpKa of 3.6). Again, calcium itself is omitted from all calculations, even those for structures solved with calcium bound. These results suggest that structural relaxation upon loss of calcium is insufficient to mitigate against pH dependence close to neutral pH, consistent with a report that intracellular acidification inhibits TMEM16A through proton competition for acidic sidechains at calcium binding sites, and that this could be a mechanism to modulate acidification that results from HCO_3_^−^ transport through the channel [39]. Further, activation of TMEM16A in the absence of intracellular calcium is modulated by voltage-dependent pH-titration, a mechanism also located to the calcium-binding pocket by mutagenesis [40].

It appears that the differential response to intracellular pH of calcium-activated chloride channels Best1 (pH-independent) and TMEM16A (pH-dependent) could be related to altered structural and electrostatic relaxation. Other reported data for TMEM16A is a reminder that proton competition with calcium is not the only or even the most common mechanism that mediates pH dependence. Extracellular protons enable activation of the channel, and widespread mutagenesis of Asp, Glu, and His residues revealed several contributions, most notably from E623 [41]. Predictions of pKa show E623 in a tightly coupled electrostatic interaction with K603, which is also predicted to be destabilised, with both being relatively buried from the solvent. The extent of conservation for TMEM16A E623 in orthologues and paralogues [41] is mirrored by that of K603. It is suggested that these amino acids could act together to generate the pH sensor that has been identified experimentally with the E623Q mutation. Interestingly, a pocket bordered by E623 and K603 sits adjacent to the ion conductance pore and is being investigated as a target for small molecule inhibition of TMEM16A [42].

### 2.4. Buried Histidine-Mediated pH Dependence in ClC-2

The ClC family consists of both voltage-gated Cl^−^ channels and Cl^−^/H^+^ exchangers, are involved in a wide range of physiological processes, and share the same basic structure, a homodimer with a chloride conductance pathway in each subunit [43] (Figure 4A), with protons modifying channel gating [44]. Family member, chloride channel ClC-2, is activated by mild acidification but inhibited by stronger extracellular acidification, with extracellular facing H532 of guinea pig ClC-2 responsible for pH-dependent channel closure [45]. In the absence of a structure for ClC-2, H532 aligns with H555 of ClC-1, for which a structure is available [46]. The Protein-Sol pKa tool gives predicted destabilising ΔpKas of 4.1 and >5 for the two H555 residues in the ClC-1 homodimer (Figure 4A,B). It appears that pH dependence is modulated by the commonly occurring mechanism of buried (from solvent, surrounded by amino acids, Figure 4B) histidine sidechain that becomes less stable as pH falls around its normal (solvent exposed) pKa value [18].

Since AlphaFold [47], via UniProt [48], supplies a confident structural model for that part of ClC-2 that maps to ordered protein in ClC-1, the monomer model was also submitted to the Protein-Sol server. Shown in Figure 4C are Protein-Sol results over the entire monomer but displayed in PyMol (with the same colour scheme as used in Protein-Sol). Amino acids predicted to be destabilised such that their pKas could contribute to pH dependence around neutral pH are indicated, showing how the Protein-Sol pKa tool can be used to conveniently screen for such groups with structures or AlphaFold models. Residue numbers are given for the AlphaFold model of human ClC-2. Thus, H530 of human ClC-2 (Figure 4C) maps to the experimental characterisation of H532 in guinea pig ClC-2 (and H555 in human ClC-1). Of the remaining four amino acids indicated, human ClC-2 E205 maps to guinea pig ClC-2 E207, also implicated in functional pH dependence [45]. This group also aligns with E148 of the *E. coli* ClC, where it is part of a proton pathway common to some other members of the ClC family [49]. The Protein-Sol pKa tool not only identifies individual amino acids that are predicted to mediate pH dependence but may also aid the discovery of groupings that form a charge/proton network.

### 2.5. Predicted pH-Sensing Residues Couple to Ligand Binding in the GABA_A_ Receptor

The major inhibitory transmitter of the adult central nervous system, γ-aminobutyric acid (GABA), is mediated primarily by the GABA_A_ receptor. GABA_A_ receptors belong to the Cys loop pentameric ligand-gated ion channel (pLGIC) family and are most commonly formed from 5 subunits in an α_1_β_2_γ_2_ 2:2:1 stoichiometry, a combination that is highly expressed synaptically and extrasynaptically in neurons [50] (Figure 5A,B). The ionotropic channels contain binding sites for modulators, including benzodiazepines and general anaesthetics [51], and are known to be strongly modulated by changes in extracellular proton concentration [52,53,54].

The human GABA_A_ α_1_β_2_γ_2_ receptor in complex with GABA and flumazenil, pdb 6d6u [55], is predicted to have electrostatically frustrated residues, α_1_H110 and β_2_D101 (both with destabilising ΔpKa > 5) in close proximity to residues β_2_Y205, α_1_F65 (aligned to α_1_F64 in mouse GABA_A_), and β_2_E155, that are known to couple with pH sensing [56,57,58] (Figure 5C). These residues are involved in GABA binding in a site formed at a subunit interface [59]. Molecular docking indicated that pH modulation of GABA_A_ may involve a combination of interactions between the binding site residues, GABA, and protons [57]. The electrostatically frustrated residues α_1_H110 and β_2_D101 surrounding the binding site could be involved in this complex mechanism through interactions involving coupled protonation and structural changes to the site (Figure 5C). Web-accessible predictions of pH-sensing centres, applied to as many structural forms that are available, can be a valuable part of the cycle to rationalise data, design experiments, and construct molecular models.

### 2.6. Sites of pH Sensitivity and Electrostatic Frustration are Readily Apparent in Glycine Receptors

Glycine receptors (GlyRs) are pentameric channels expressed in the central nervous system where they play essential roles in mediating inhibitory neurotransmission [60]. GlyRs are modulated by extracellular pH changes, which occur under physiological conditions. Both α1 homomeric and α1β heteromeric GlyR pentamers are inhibited by acidic extracellular pH [61,62]. Site-directed mutagenesis showed that H109 and T112 in α1 subunits are key residues for proton modulation of GlyRs [62]. H109 is a highly conserved residue across all α GlyR subunits and exhibits the largest destabilising ΔpKa (>5) in calculations with the human α3 GlyR homopentamer (5fcb, Figure 6) [63]. It is buried at an interface with a neighbouring subunit that also involves T112 so that protonation and conformational change are likely to be coupled.

Two groups with lower predicted electrostatic frustration than H109 are R271 and R252 (Figure 6). Arginine residues (with high standard pKa) are not normally directly involved in neutral pH dependence, but in the case of R271, electrostatic repulsion is thought to underlie mutation that causes hereditary hyperekplexia [64]. An electrostatically frustrated environment is immediately apparent in the predictions from Protein-Sol, with repulsion also following from the proximity of symmetry-related arginine amino acids. This occurs against a background of ionisable groups that are mostly stabilised in their neutral pH forms (or at least not substantially destabilised).

## 3. Discussion

Mutagenesis screens for sites of pH dependence in biological processes can be complicated by the degree to which conformational and interface changes couple to the pH dependence [18]. At the centre, though, will be groups changing protonation state over the process, and thus user-friendly visualisation of reliable predictions for such groups will be useful in guiding experimental design. One such tool (www.protein-sol.manchester.ac.uk/pka) [20] is used in this work. Using the example of human Best1, a calcium-activated chloride channel, it is shown that not only does the catalogue of disease-causing mutations segregate by proximity to the calcium-binding site, but also that site extends beyond the immediate carboxylate ligands to include a region that has evolved a disorder/order transition alongside calcium binding. This transition is predicted to decouple calcium and proton binding (at neutral pH), consistent with an observed lack of functional pH dependence [21]. On the other hand, the persistence of predicted electrostatic frustration for calcium ligands of the TMEM16A calcium-activated chloride channel suggests a pH dependence that has been reported experimentally [39].

Acid-induced inhibition of TMEM16A may be a negative feedback mechanism to prevent damaging acidification as a result of HCO_3_^−^ transport through the channel. The Best2 channel can conduct HCO_3_^−^ ions [65] and four human Best homologues (hBest1, hBest2, hBest3, and hBest4) are highly permeable to HCO_3_^−^ suggesting a role in maintaining HCO_3_^−^ homeostasis [25]. Mechanisms of pH regulation in the RPE, where Best1 is strongly expressed, involve several Na+-dependent HCO_3_^−^ and HCO_3_^−^/Cl^−^ transporters and the basolateral ClC-2 Cl^−^ channel, which is inhibited by strong extracellular acidification. There may be no need for a pH dependence of Best1 if the epithelium pH is regulated by these channels.

A parallel distinction between pH-dependent and pH-independent calcium binding and function is discussed for the well-studied pair of cytosolic proteins, calmodulin and calbindin D9k, also flagged by a different extent of change in predicted electrostatic frustration between calcium bound and unbound forms. The suggestion that structure-based visualisation of electrostatic frustration, with a simple web tool, can highlight sites of functional interest and aid in hypothesis forming is apparent for the calcium-binding protein family. The next steps in that area will be more extensive searches of calcium-binding sites, followed by biophysical and functional studies of calcium/proton coupling in select systems. Given the extent of disease mutation clustering around the calcium site in hBest1, it may also be fruitful to analyse disease-associated mutations in other calcium-dependent systems. One area to look at is voltage-gated calcium channels, for which functionally important pH dependence of activity is known [66], arising from proton competition at acidic calcium-binding sites [67].

The other channels studied in the current work (ClC-2, GABA_A_, GlyR), completing representation across the human chloride families, reinforce the message that buried histidine sidechains are common mediators of pH dependence, although the precise balance of charge interactions (destabilising dehydration and potentially stabilising charge-charge) will affect the pKa and degree of electrostatic frustration. Elevated pKas for aspartic and glutamic acid pKas may also be involved, predicted to couple with histidine sidechain, ligand binding, and conformational change. Predicted centres of proton titration around neutral pH may also indicate pathways for experimental study. A parallel approach has been demonstrated in studies of proton-sensing G protein-coupled receptors [68]. Importantly, the advent of confidently predicted AlphaFold models covering much of the human proteome [47] allows facile extension to many systems and human diseases and, through hypothesis testing, potential clinical relevance.

## 4. Materials and Methods

### pKa Calculations and Visualisation

The Protein-Sol pKa calculation server (www.protein-sol.manchester.ac.uk/pka) [20] was used to predict pKas for ionisable amino acid sidechains (Asp, Glu, Lys, Arg, His) in a sphere around each chosen centre. These calculations can be accumulated around multiple centres so as to cover larger regions of a protein or complex of proteins. Results are visualised on the server with colour coding according to predicted pKa change, using the B-factor field of a PDB coordinate file. In order to use the dynamic range of the fixed format B-factor field, ΔpKas are multiplied by 10. Additionally, the sign of a ΔpKa is adjusted such that a negative value maps to destabilisation of the ionised form, and a positive value maps to stabilisation. A user can then conveniently view stabilising and destabilising groups through colour coding (destabilising red, and stabilising blue). Scaled ΔpKa values are limited to −50 minimum and 50 maximum (mapping to actual ΔpKa range from −5 to +5). A protein surface salt bridge typically gives a stabilising ΔpKa that does not exceed 1 (10 for the scaled value) so that any values at the minimum or maximum limits are very substantially stabilised or destabilised. The Protein-Sol pKa server is based on a combined finite difference Poisson−Boltzmann (FDPB)/Debye−Hückel (DH) method for predicting pKas, termed FDDH, which has been validated against experimental data for solvent-exposed and more buried ionisable groups. ΔpKa values are with reference to model compound pKas, which in this method are 4.0 for aspartic acid, 4.4 for glutamic acid, 6.3 for histidine, 10.4 for lysine, and 12.0 for arginine [22]. The server was developed for ease of visualisation of groups that are electrostatically frustrated (typically at neutral pH) and are therefore, candidate pH sensing groups. It was benchmarked, for identifying electrostatically frustrated ionisable sites, with functional groups from influenza virus hemagglutinin and a designed pH-sensor protein [20]. An additional method was employed to calculate pKas, in order to compare with FDDH results. The empirical calculation tool propka [24] implemented in pdb2pqr at the APBS server [69] was used (with default parameters). Protein structures were retrieved from the Protein Data Bank (PDB) [70], with additional visualisation in PyMol and Swiss PDB Viewer [71].

## Figures and Tables

**Figure 1 molecules-28-05753-f001:**
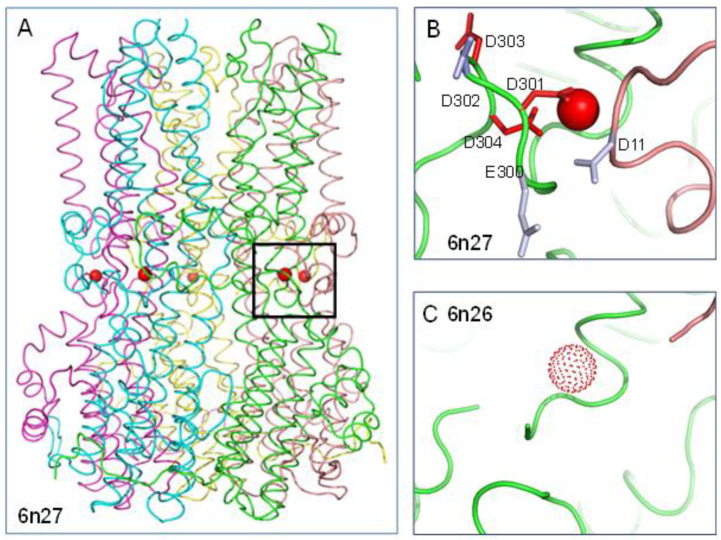
Electrostatic environment of the cBest1 calcium clasp. (**A**) Calcium clasp location in cBest1. (**B**) Calcium ion location (red sphere) and colour-coded representation for ΔpKas of pH-titratable sidechains adjacent to the calcium site in the holo structure (6n27). Although calcium is displayed, pKa calculations are made without it, to enable direct comparison with the apo structure. Red sidechains denote destabilisation of the ionised form and electrostatic frustration, and blue are stabilised. (**C**) The calcium-binding loop is disordered in the apo structure (6n26), with the calcium ion location copied over from the holo form for reference.

**Figure 2 molecules-28-05753-f002:**
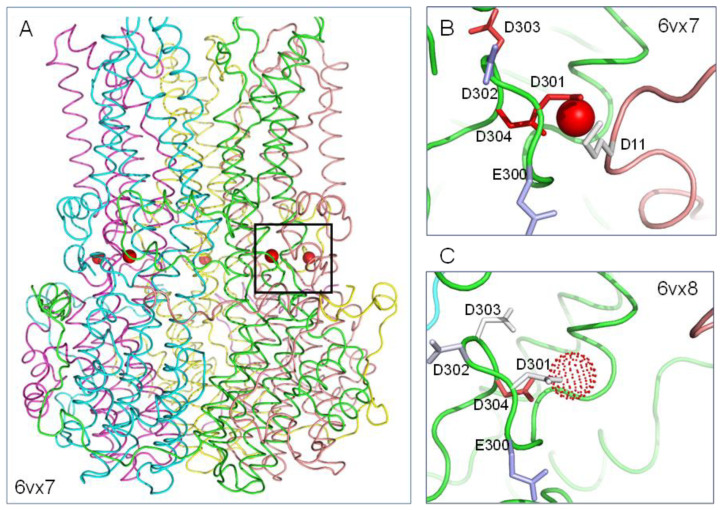
Electrostatic environment of the bBest2 calcium clasp. (**A**) Clasp location in bBest2. (**B**) pH-titratable sidechain colour coding, and ΔpKa calculations as for Figure 1, in the holo structure (6vx7). (**C**) Structural relaxation of the calcium-binding region and reduction of carboxylate sidechain electrostatic frustration in the apo form (6vx8).

**Figure 3 molecules-28-05753-f003:**
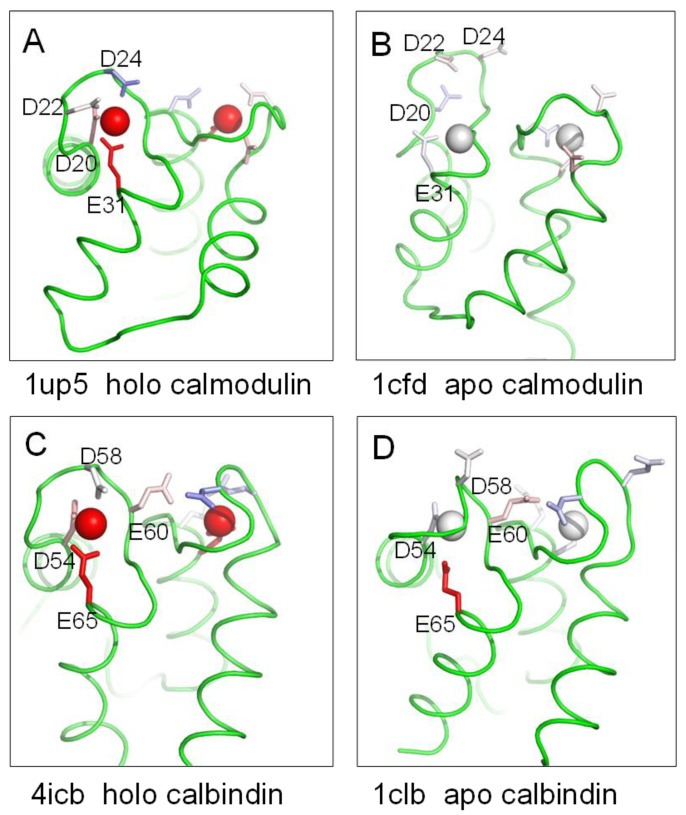
Electrostatic environment of calcium binding sites in calmodulin and calbindin D_9k_. Two calcium (EF-hand) sites are shown in each plot, with ionisable group colour-coding from the Protein-Sol pKa tool (see Figure 1). Sidechain residue labels are only given for one of the two sites, in each case. (**A**) Holo calmodulin 1up5 [30]. (**B**) Apo calmodulin 1cfd [26]. (**C**) Holo calbindin 4icb [31]. (**D**) Apo calbindin 1clb [32].

**Figure 4 molecules-28-05753-f004:**
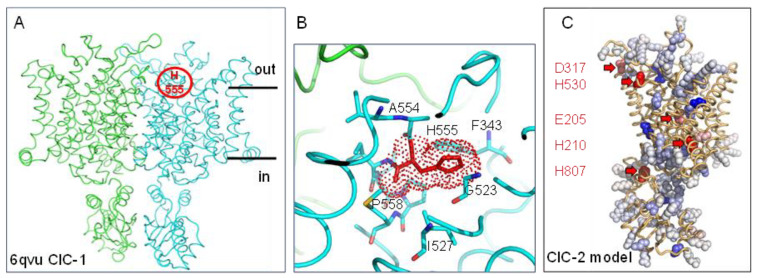
Sites of pH sensing in the ClC family. (**A**) Cartoon tube representation of human ClC-1 dimer structure (pdb 6qvu [46]), with the location of H555 indicated in one monomer. (**B**) An expanded view of the packing environment around the H555 pH sensor, with groups labelled, and a destabilising pKa from Protein-Sol denoted (red). (**C**) Protein-Sol output is visualised (PyMol) for a monomer AlphaFold model of human ClC-2. Groups predicted to mediate pH-dependent effects around neutral pH are highlighted and labelled.

**Figure 5 molecules-28-05753-f005:**
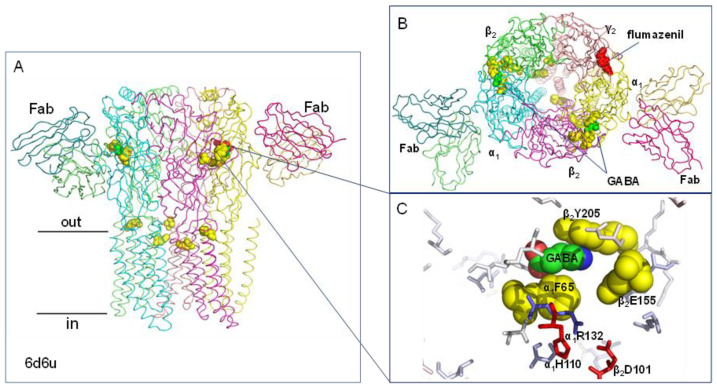
Charge cluster around the GABA binding site of the GABA_A_ receptor. (**A**) Side view (along the membrane) of GABA_A_ receptor with bound Fab fragments labelled, structure pdb 6d6u, [55]. Colouring is by chain with residues linked to pH sensing shown as yellow spheres, GABA as green spheres and flumazenil as red spheres. (**B**) Top view (into the membrane) with subunits labelled. (**C**) GABA binding site (green spheres) with residues linked to pH sensing shown as yellow spheres and residues predicted to be electrostatically frustrated residues as red sticks.

**Figure 6 molecules-28-05753-f006:**
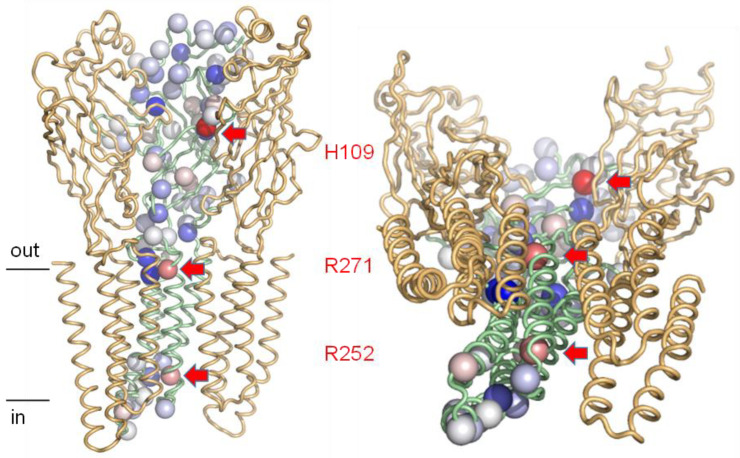
Electrostatic frustration in GlyR (5cfb) [63] identifies functional sites of pH sensing and electrostatic repulsion. Protein-Sol calculated and colour-coded ΔpKas (red destabilising, blue stabilising) are illustrated for a single protomer of the pentamer with spheres on ionisable group Cα atoms. Two neighbouring monomers are also displayed (cartoon representation), leaving a ‘cut-away’ view of the channel, which is displayed along the membrane (left-hand side) and with a tilted view (from the intracellular side) to indicate the channel pore. Groups predicted to be electrostatically frustrated are labelled and indicated with arrows on the left and righthand plots.

## Data Availability

The datasets used and analysed during the current study are available from the corresponding author upon reasonable request.

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
