# Peer review of "Computational Investigation of Mechanisms for pH Modulation of Human Chloride Channels"

_molecules, 2023, doi:10.3390/molecules28155753_

Round 1

Reviewer 1 Report

The study utilizes a previously developed web-based method for calculation of pKa values and quick display of amino acids with shifted pKa values. This tool is applied to study the pH dependence of several chloride channels. Some channels they discuss are calcium activated and some are not. For calcium activated channels they find that the pH independent channel structure relaxes and pKa values normalize when calcium is removed from the structure while for the pH dependent channel that is not the case. They compare this to the well studied example of calmodulin (pH independent) which relaxes more and calbindin D9k (pH dependent) which relaxes less, and they discuss the hypothesis that the extent of disordering and/or structural rearrangement in the absence of calcium binding modulates pH-dependence. I find this hypothesis interesting and worthy of publication.  Finally, they discuss pH dependence of additional chloride channels, some of which are calcium modulated and some are not.

I would like to see the following changes:

1) Better define and explain what electrostatic frustration is. This term is used throughout the paper, but its explanation in the paper is just: destabilized ionisable charge sites. By looking up other publications it looks to me that frustrated sites are sites experiencing pKa shifts which favor protonation states other than the protonation states in water. How does electrostatic frustation dependent on pH of interest and what is the relationship between the calculated pKa and pH of interest. Say if an Asp pKa is 5.5 or 6.5 or 7.5 and pH is 7.4, which ones of those sites are frustrated? If the magnitude of the shift matters and pH is important, how are electrostatically frustrated sites identified through only the color schemes in the figures?

2) Calculated pKa values are important and should be included in the paper.

3) I would like to see the logic of the paper reorganized somehow, particularly the introduction. When I started reading the introduction I thought that the paper is about Bestropins. Then there was significant amount of time spent on ASICs, which then didn't show up later in the paper. My understanding is that this paper is mostly about concepts, and that should be emphasized in the introduction.

In the results, I think it would be good to see a division of calcium modulated channels vs those that are not.

Also regarding the logic, try not giving out details that are not necessary, as they are distracting from the big picture. Or include them in a table or SI.

4) For each protein case discussed here, clearly point out experimental data and how do the predictions compare to those.

5) As far as I understand, the advantage of protein-sol is that it provides instant visualization of sites with shifted pKa values. How does the accuracy of prediction compare to other widely used pKa prediction methods, say propka? Is there a bechmark for the accuracy? It would be interesting to see the comparison for the calculated pKa values with say propka.

Author Response

Reviewer 1

1) Better define and explain what electrostatic frustration is. This term is used throughout the paper, but its explanation in the paper is just: destabilized ionisable charge sites. By looking up other publications it looks to me that frustrated sites are sites experiencing pKa shifts which favor protonation states other than the protonation states in water. How does electrostatic frustation dependent on pH of interest and what is the relationship between the calculated pKa and pH of interest. Say if an Asp pKa is 5.5 or 6.5 or 7.5 and pH is 7.4, which ones of those sites are frustrated? If the magnitude of the shift matters and pH is important, how are electrostatically frustrated sites identified through only the color schemes in the figures?

Response: Thank you for the suggestion.  Alongside a reduction of Introduction content on bestrophin mutations and acid sensing ion channels (reduced text lines 62 – 66 and 75 - 77), we have added information (lines 88 – 98) on the term electrostatic frustration, how it relates to destabilisation of buried ionisable groups, and how predictions of this frustration are viewed on the protein-sol web server.  We discuss that the predicted pKa shift from model compound values determines the degree of stabilisation or destabilisation (electrostatic frustration).  In the Reviewer’s example of Asp any pKa above the model compound value (around 4.4), is destabilised, the greater the upward pKa shift, the greater the destabilisation.

2) Calculated pKa values are important and should be included in the paper.

Response: The Reviewer is referring to our only listing pKa shifts in the text for some of the systems, and otherwise relying on the colour-coding of amino acids in figures. In response we have added a note at first quotation of pKa shifts in the results section (lines 126 – 128), that these shifts (and the corresponding colour ramping) are restricted to a magnitude of 5, which is sufficient to move aspartic and glutamic carboxylate sidechain pKas to up above neutral (and physiological) pH. This is already discussed in the Methods section. We also update notation so that any DpKa capped at a magnitude of 5 is written as >5 (lines – throughout), and described as stabilising or destabilising. We have now added text so that predicted pKa shifts are present in all Results sections.

3) I would like to see the logic of the paper reorganized somehow, particularly the introduction. When I started reading the introduction I thought that the paper is about Bestropins. Then there was significant amount of time spent on ASICs, which then didn't show up later in the paper. My understanding is that this paper is mostly about concepts, and that should be emphasized in the introduction.

In the results, I think it would be good to see a division of calcium modulated channels vs those that are not.

Also regarding the logic, try not giving out details that are not necessary, as they are distracting from the big picture. Or include them in a table or SI.

Response: In response to these comments on overall structure and flow, we have:

  • Reduced the discussion of bestrophin (lines 62 – 66) and removed the original results section describing the location of bestrophin disease mutations (see also Reviewer 3).
  • Focussed the discussion of ASICs (in the Introduction) on the general nature of structural relaxation and pH-dependence in carboxylate-mediated calcium binding sites (from line 74).
  • Added discussion on the concepts (electrostatic frustration and pKas) in the Introduction (lines 88 – 98).
  • Regarding the ordering of chloride channels that do (Results sections 2.1, 2.3), or do not (Results sections 2.4, 2.5, 2.6), bind calcium. These are sequential, with the intervening section 2.2 on calmodulin and calbindin. We feel that it is fair to have these calcium-binding proteins in 2.2, as a cytosolic counterpoint to the bestrophin example of 2.1, but also exhibiting variation in the extent of coupling between proton and calcium binding.
  • With regard to details, In addressing other points we have removed some detail (e.g. discussion of BVMD and ARB mutations, and acid sensing ion channels). Remaining information, e.g. heading results sections, we feel is valuable to the reader for some physiological context of each system.

4) For each protein case discussed here, clearly point out experimental data and how do the predictions compare to those.

Response: This is an interesting point.  For most of the systems studied here, and for the targets of this work, experimental observations derive from physiological read-outs upon mutation, as opposed to precise measurements of pKa.  We have included the literature information we have found, largely reports of pH-dependence or pH-independence and the results of mutation studies, and those form the core of our comparison with experiment.

5) As far as I understand, the advantage of protein-sol is that it provides instant visualization of sites with shifted pKa values. How does the accuracy of prediction compare to other widely used pKa prediction methods, say propka? Is there a bechmark for the accuracy? It would be interesting to see the comparison for the calculated pKa values with say propka.

Response: With regard to the first part of this comment, the Reviewer is right that the pka tool at our protein-sol server provides prediction and visualisation of predicted pKas, we have expanded on this aspect in the Introduction (also point 1 of this Reviewer). For benchmarking we have added text in the Materials and Methods section noting that the original report of the online pKa tool in 2020, successfully identified functional ionisable group sites in influenza virus hemagglutinin and a designed pH sensor (lines 430 – 432). Further comparison with propka is an excellent suggestion, that we have already accommodated for bestrophin, in Results section 2.1.

Reviewer 2 Report

The paper is clearly written, in very good English. The methods are adequately described and cited. The results are presented in detail and all findings are discussed and supported successfully.

I recommend that this paper be accepted for publication.

Author Response

Reviewer 2

The paper is clearly written, in very good English. The methods are adequately described and cited. The results are presented in detail and all findings are discussed and supported successfully. 

I recommend that this paper be accepted for publication.

Response: We thank this Reviewer for their support.

Reviewer 3 Report

The manuscript by Elverson et al. tackles the question of pH regulation of various chloride channel types, and the interplay between H+ and Ca2+ ions for these and the Ca2+ sensor proteins calmodulin and calbindin. Two core ideas are presented: (1) that a mechanism to confer sensitivity to Ca2+ but not H+ ions is to have relaxed/disordered structures in the absence of Ca2+ that lower the pKa values of protonatable residues (glutamate and aspartate) to levels well below physiological pH, and (2) that electrostatically frustrated residues, identifiable with the program Protein-sol pKa, are likely candidates for proton modulation of various types of chloride ion channels. Overall, I find the presented ideas intriguing, and Protein-sol pKA as an exciting tool for exploring proton modulation of ion channels, as well as membrane receptors. For example, GPCRs which were recently shown to be broadly regulated by protons (e.g., https://pubmed.ncbi.nlm.nih.gov/33478938/; https://www.pnas.org/doi/10.1073/pnas.2100171118).   

I do have some suggestions that I think would help improve the clarity of the presented ideas and the overall readability of the manuscript:

1. The annotation of BVMD and ARB mutations within the hBest1 channel structure seems out of place, with very little connection with the main hypotheses outside of a suggested link between mutations and Ca2+ sensitivity in lines 197 to 203. I suggest removing this section as it distracts from the main subjects of the paper.

2. The abstract would be more effective if the aims and core hypotheses were stated more clearly. For example, I find the following statement confusing: “This study shows how web-based calculation of pKas allows rapid identification of amino acids of interest with respect to pH-dependence”. Firstly, this statement seems more applicable to the author’s previous publication when they first described this program. Secondly, this is not one of the core questions being addressed in the paper. Rather, the program was used to address the two core questions noted above. Indeed, I think the abstract could be rewritten to focus on the core ideas, and referring to Protein-sol pKa as a tool used to test these ideas.

3. Related to point 1 above, removal of lines 63-70 (about Bestrophin mutations) would help shorten the introduction. Furthermore, I found it difficult to hone in on the central objectives within the introduction (i.e., how do the following integrate into a series of objectives with a common narrative?  i) mutations, ii) structural relaxation for Ca2+/H+ dynamics, iii) electrostatically frustrated residues as proton sensors, and iv) implementation of Protein-sol pKa as a tool to ID such residues). It would be useful if the authors could edit/rewrite the introduction to better delineate these various ideas and integrate them whenever possible.

4. For those who are not familiar with ASIC channels, I think the description of the ring of glutamate residues recently shown to mediate Ca2+ sensitivity as well as H+ sensitivity could be misinterpreted. Specifically, without a description of the central molecular determinants for proton activation of ASIC channels (i.e., H79 in the wrist region, K211 in the palm, and the acid pocket), this statement implies that the noted ring structure is central to proton activation, which to the best of my knowledge is not the case.

5. Lines 264-277 seem out of place and perhaps more appropriate for the discussion section.

6. Although this manuscript focuses strictly on chloride channels and Ca2+-sensor proteins, it might be relevant and interesting for the authors to add a short section in their discussion about the selectivity filters of voltage-gated calcium channels, given that these are comprised of protonatable glutamate and aspirated residues that form a high affinity binding site for Ca2+. There is substantial literature on the effects that protons have on Ca2+ permeation though calcium channels, and evidence that protons interact with the same acidic residues in the pore that are required for Ca2+ selectivity.

Author Response

Reviewer 3

Overall, I find the presented ideas intriguing, and Protein-sol pKA as an exciting tool for exploring proton modulation of ion channels, as well as membrane receptors. For example, GPCRs which were recently shown to be broadly regulated by protons (e.g., https://pubmed.ncbi.nlm.nih.gov/33478938/; https://www.pnas.org/doi/10.1073/pnas.2100171118).   

Response: This is an interesting and directly relevant publication, which we have added to the Discussion session (lines 405 – 407).

I do have some suggestions that I think would help improve the clarity of the presented ideas and the overall readability of the manuscript:

  1. The annotation of BVMD and ARB mutations within the hBest1 channel structure seems out of place, with very little connection with the main hypotheses outside of a suggested link between mutations and Ca2+ sensitivity in lines 197 to 203. I suggest removing this section as it distracts from the main subjects of the paper.

Response: We accept the Reviewer’s reasoning and have removed BVMD and ARB mutation sections, i.e. original Results 2.1 removed, with subsequent sections renumbered, and also Materials and Methods original section 4.1 removed, also covering the mutations modelling and visualisation. The relevant (mutations) figure (original Fig. 1) has also been removed, so that original figures 2 to 7 become figures 1 to 6.  Discussion of BVMD and ARB mutations has also been reduced in the Introduction (lines 62 – 66).

  1. The abstract would be more effective if the aims and core hypotheses were stated more clearly. For example, I find the following statement confusing: “This study shows how web-based calculation of pKas allows rapid identification of amino acids of interest with respect to pH-dependence”. Firstly, this statement seems more applicable to the author’s previous publication when they first described this program. Secondly, this is not one of the core questions being addressed in the paper. Rather, the program was used to address the two core questions noted above. Indeed, I think the abstract could be rewritten to focus on the core ideas, and referring to Protein-sol pKa as a tool used to test these ideas.

Response: To address this concern, we have changed the emphasis in the Abstract near beginning and near the end, maintaining the point that pKa predictions allow hypotheses (which are detailed in the Abstract) to be generated, but not making those methods the focus of the Abstract (lines 15 – 16, 26).

  1. Related to point 1 above, removal of lines 63-70 (about Bestrophin mutations) would help shorten the introduction. Furthermore, I found it difficult to hone in on the central objectives within the introduction (i.e., how do the following integrate into a series of objectives with a common narrative?  i) mutations, ii) structural relaxation for Ca2+/H+ dynamics, iii) electrostatically frustrated residues as proton sensors, and iv) implementation of Protein-sol pKa as a tool to ID such residues). It would be useful if the authors could edit/rewrite the introduction to better delineate these various ideas and integrate them whenever possible.

Response: Reviewer 3 makes a very similar point to that of Reviewer 1 (point 3), so that our response repeats that to Reviewer 1, with substantial changes made to flow in the Introduction:

  • Reduced the discussion of bestrophin (lines 62 – 66) and removed the original results section describing the location of bestrophin disease mutations.
  • Focussed the discussion of ASICs (in the Introduction) on the general nature of structural relaxation and pH-dependence in carboxylate-mediated calcium binding sites (from line 74).
  • Added discussion on the concepts (electrostatic frustration and pKas) in the Introduction (lines 88 – 98).
  1. For those who are not familiar with ASIC channels, I think the description of the ring of glutamate residues recently shown to mediate Ca2+ sensitivity as well as H+ sensitivity could be misinterpreted. Specifically, without a description of the central molecular determinants for proton activation of ASIC channels (i.e., H79 in the wrist region, K211 in the palm, and the acid pocket), this statement implies that the noted ring structure is central to proton activation, which to the best of my knowledge is not the case.

Response: We recognise that there has been much work on the structure and function of acid sensing ion channels. Whilst the original reference we give (Zuo et al 2018) gives clear evidence (through mutation) of the role of a ring of glutamate residues modulating pH-dependence, we have now also referred to a 2020 review (Rook et al 2021, ref 16, lines 75 - 77) that gives a wider view on residues and regions that may be involved in pH-dependence of ASICs.

  1. Lines 264-277 seem out of place and perhaps more appropriate for the discussion section.

Response: We have looked at this and see the Reviewer’s point but we feel that on balance, with some reporting of calculated results in this paragraph, it would be better as remaining in the Results section.

  1. Although this manuscript focuses strictly on chloride channels and Ca2+-sensor proteins, it might be relevant and interesting for the authors to add a short section in their discussion about the selectivity filters of voltage-gated calcium channels, given that these are comprised of protonatable glutamate and aspirated residues that form a high affinity binding site for Ca2+. There is substantial literature on the effects that protons have on Ca2+ permeation though calcium channels, and evidence that protons interact with the same acidic residues in the pore that are required for Ca2+ selectivity.

Response: Thank you for this suggestion, interesting and entirely appropriate, and we have added a note on voltage-gated calcium channels in the Discussion section (lines 395 – 297, new references 66 and 67).

Round 2

Reviewer 1 Report

Most of my comments were addressed. I would still like to have the pKa values published. It is OK if they are published as supplementary info. 

Author Response

We are glad to have addressed all the Reviewers' issues (except one) to their satisfaction. That single issue, from Reviewer 1, is that they would still like to see pKa values listed.

Response: We thought that we had addressed this in our first revision, with listing of all ΔpKa values for each amino acid sidechain being considered.  In retrospect though we see the Reviewer’s point, we need the model compound pKas, so that a reader knows how to derive predicted pKa (from model compound pKa + ΔpKa).  To correct this, we have added two small sections of text (in red), in Materials and Methods we list the model compound pKas for ionisable groups studied, and then at first presentation of predicted ΔpKas (results section 2.1) we list both predicted ΔpKas and pKas.  Subsequently we list only predicted ΔpKas, having established their relationship to pKas, and since they give a greater insight to the relevant feature (stabilisation or destabilisation at a site) than the pKa itself.

Sorry for the delay in making this small revision, we hope is satisfactory.